# Chemical Profile, Antioxidant Properties and Antimicrobial Activities of Malaysian *Heterotrigona itama* Bee Bread

**DOI:** 10.3390/molecules26164943

**Published:** 2021-08-15

**Authors:** Joseph Bagi Suleiman, Mahaneem Mohamed, Ainul Bahiyah Abu Bakar, Victor Udo Nna, Zaida Zakaria, Zaidatul Akmal Othman, Abdulqudus Bola Aroyehun

**Affiliations:** 1Department of Physiology, School of Medical Sciences, Universiti Sains Malaysia, Kubang Kerian 16150, Kelantan, Malaysia; bagisuleiman@yahoo.com (J.B.S.); ainul@usm.my (A.B.A.B.); zaida_zakaria@ymail.com (Z.Z.); drzaida87@gmail.com (Z.A.O.); 2Department of Science Laboratory Technology, Akanu Ibiam Federal Polytechnic, Unwana P.M.B. 1007, Ebonyi State, Nigeria; 3Unit of Integrative Medicine, School of Medical Sciences, Universiti Sains Malaysia, Kubang Kerian 16150, Kelantan, Malaysia; 4Department of Physiology, Faculty of Basic Medical Sciences, College of Medical Sciences, University of Calabar, Calabar P.M.B. 1115, Cross River State, Nigeria; victor2nna@gmail.com; 5Unit of Physiology, Faculty of Medicine, Universiti Sultan Zainal Abidin, Kuala Terengganu 20400, Terengganu, Malaysia; 6Nutrition and Dietetics Program, School of Health Sciences, Universiti Sains Malaysia, Kubang Kerian 16150, Kelantan, Malaysia; bqaroyehun@student.usm.my; 7Clinical Nutrition, The Rowett Institute of Nutrition and Health, University of Aberdeen, Scotland AB24 3FX, UK

**Keywords:** *Heterotrigona itama* bee bread, polyphenols, microbial activities

## Abstract

The aim of the study was to determine the chemical profile, antioxidant properties and antimicrobial activities of *Heterotrigona itama* bee bread from Malaysia. The pH, presence of phytochemicals, antioxidant properties, total phenolic content (TPC) and total flavonoid content (TFC), as well as antimicrobial activities, were assessed. Results revealed a decrease in the pH of bee bread water extract (BBW) relative to bee bread ethanolic extract (BBE) and bee bread hot water extract (BBH). Further, alkaloids, flavonoids, phenols, tannins, saponins, terpenoids, resins, glycosides and xanthoproteins were detected in BBW, BBH and BBE. Also, significant decreases in TPC, TFC, DPPH activity and FRAP were detected in BBW relative to BBH and BBE. We detected phenolic acids such as gallic acid, caffeic acid, trans-ferulic acid, trans 3-hydroxycinnamic acid and 2-hydroxycinnamic acid, and flavonoids such as quercetin, kaempferol, apigenin and mangiferin in BBE using high-performance liquid chromatography analysis. The strongest antimicrobial activity was observed in *Klebsilla pneumonia* (MIC_50_ 1.914 µg/mL), followed by *E. coli* (MIC_50_ 1.923 µg/mL), *Shigella* (MIC_50_ 1.813 µg/mL) and *Salmonella typhi* (MIC_50_ 1.617 µg/mL). Bee bread samples possess antioxidant and antimicrobial properties. Bee bread contains phenolic acids and flavonoids, and could be beneficial in the management and treatment of metabolic diseases.

## 1. Introduction

Bee bread comprises of pellets of bee pollen packed by foraging bees and conveyed to the bee hive on their hind legs in a pollen bag [1]. It is made from pollen assembled by bee and mixed with honey and salivary enzymes. Bee bread contains a high amount of lactic acid relative to bee pollen and also contains protein, antioxidant and phenolic compounds [2].

The scavenging worker bees are responsible for the collection of pollen pellets from flowers back to the hives, which are then assembled by younger bees into compartments of the brood comb [3]. Furthermore, honey is placed on the pollen to avoid decay, after which microorganisms like bacteria/yeast act upon it to produce bee bread in a process referred to as lactic acid fermentation. Thereafter, it is eaten by adult bees and also used as a source of food to feed larvae [4]. Generally, bees obtain protein from pollen which are separated from nectar. Bee bread is composed of 20% protein, 24–35% carbohydrates, 3% lipids, 3% minerals and vitamins (B1, B2, C, E, K, biotin, folic acid and nicotinic acid), pantothenic acid, polyphenols (phenolic acid and flavonoids), sterols, enzymes (amylase, saccharase and phosphatases) and carotenoids [5,6].

It has been suggested that bee bread possesses a large amount of protein and amino acid because bee pollen also possesses a high amount of these nutrients [7]. It has also been found that the amount of protein in bee bread depends largely on the type of flower, pollen grains and region where they are located. Considerable amounts of enzymes (phosphatase, glucose-oxidase and amylase) and amino acids (glutamic acid, proline, aspartic acid, arginine, histidine, valine, leucine, isoleucine, methionine, lysine, tryptophan, threonine, phenylalanine, cysteine, alanine, tyrosine, glycine and serine) are present in bee bread [8].

The hydrolysis of monosaccharides during lactic acid fermentation of the pollen produces 0.12% sucrose, 3.37% maltose, 42.59% glucose and 57.51% fructose, which are present in bee bread, and 1.82% disaccharides (trehalose, turanoe and isomaltose) are also found as well [9]. The process of fermentation where *Lactobacillus* sp. uses carbohydrates as a source of oxygen to produce lactic acid up to a concentration of 3.2% occurs in the hives [8].

Kaplan et al. [10] reported that eight varieties of Turkish bee bread contained 1.9–2.54% ash, 14.8–24.3% protein, 5.9–11.5% fat, and 11.4–15.9% moisture, as well as fatty acids. Apart from that, bee bread contains magnesium, calcium, phosphorus, sodium, iron, potassium, zinc, manganese, copper, silicon and aluminium [11]; vitamins (folic acid, niacin, vitamin B6, vitamin C, pantothenic acid), and flavonoids (pinocembrin) [12].

Studies on varieties of Morroccan bee bread to prove its resistance to antibiotics revealed strong antimicrobial activities on the bacterial strains [13]. On the other hand, a research study carried out by Tichy and Novak [14] revealed that bee bread hydrophobic compounds had antibacterial activity against *Viridans streptococci*. Bee bread was also found to possess antibacterial activity against *Staphlococcus aureus* and *Staphlococcus epidermidis* [15].

The antifungal and antioxidant activities of bee bread and their relationship with phenolic compounds have been reported [13]. Similarly, studies have also revealed anti-inflammatory activities of bee bread, making it one of the strongest anti-inflammatory natural products [16,17,18]. Furthermore, antitumor effects of bee bread have been reported on ovary, hepatoma, prostate, bladder, melanoma and renal cancer cells, with varrying mechanisms of action depending on the tumor type [19]. Immunoactivating activity of bee bread has been reported in different studies [13,20]. The aim of this present study was to assess and compare the phytochemical screening, antioxidant properties, high-performance liquid chromatography and antimicrobial activities of extracts of Malaysian *Heterotrigona itama* bee bread.

## 2. Results

### 2.1. pH of Heterotrigona itama Bee Bread

There was a significant decrease (*p* < 0.05) in the pH of *Heterotrigona itama* bee bread water extract (BBW) relative to *Heterotrigona itama* bee bread ethanolic extract (BBE). The pH did not vary significantly between BBE and *Heterotrigona itama* bee bread hot water extract (BBH) (Figure 1).

### 2.2. Phytochemical Screening Analysis of Heterotrigona itama Bee Bread

Qualitative data of phytochemicals revealed an abundance of alkaloids, flavonoids, phenols, terpenoids, resins and glycosides in BBE compared to BBW and BBH, while similar results were obtained in all three samples for tannins, saponins and xanthoproteins. In BBE, the most abundant phytochemicals were terpenoids and flavonoids (Table 1).

### 2.3. Analysis of In Vitro Antioxidant Assessment of Heterotrigona itama Bee Bread

The in vitro antioxidant assessments were carried out on BBE, BBW and BBH using 1,1-diphenyl-2-picrylhydrazyl (DPPH) radical, hydrogen peroxide (H_2_O_2_) scavenging, total phenolic and flavonoid contents, as well as ferric ion reducing antioxidant power (FRAP) assays.

#### 2.3.1. DPPH Radical and H_2_O_2_ Scavenging Activities of *Heterotrigona itama* Bee Bread

The DPPH radical scavenging activity of BBE was significantly higher than BBW and BBH. However, BBH was significantly higher than BBW (Figure 2a,b). The half-maximal inhibitory concentration (IC_50_) of BBE and BBW were 2.139 µg/mL and 2.694 µg/mL, respectively. Interestingly, the IC_50_ for BBH was slightly lower (1.745 µg/mL) than BBE and BBW (Figure 2a,b).

Similarly, the scavenging activity of H_2_O_2_ showed that BBW and BBH were significantly lower relative to BBE (Table 2). While the IC_50_ of BBW and BBH were 1.855 µg/mL and 1.945 µg/mL, respectively, the IC_50_ for BBE was lower (1.338 µg/mL) (Figure 3a,b).

#### 2.3.2. Total Phenolic and Flavonoid Contents of *Heterotrigona*
*itama* Bee Bread

The total phenolic and flavonoid contents were significantly higher in BBE relative to BBW and BBH. However, the flavonoid content was significantly higher in BBH relative to BBW. BBE had the highest phenolic and flavonoid contents (Table 2).

#### 2.3.3. Ferric Ion-Reducing Antioxidant Power (FRAP) of *Heterotrigona*
*itama* Bee Bread

FRAP activity was significantly lower in BBW and BBH relative to BBE. The lowest FRAP activity was detected in BBH, and was significantly lower relative to BBW. On the other hand, BBE had the highest FRAP activity (Table 2).

#### 2.3.4. Phenolic Compounds Analysis of BBE Using High-Performance Liquid Chromatography (HPLC)

Nine phenolic compounds were identified in BBE using HPLC analysis. These include phenolic acids such as gallic acid, caffeic acid, trans-ferulic acid, trans 3-hydroxycinnamic acid and 2-hydroxycinnamic acid, and flavonoids such as quercetin, kaempferol, apigenin and mangiferin. The result in the chromatogram revealed the retention time for compounds from the least to the highest retention time: gallic acid < caffeic acid < mangiferin < trans-ferulic acid < trans 3-hydroxycinnam acid < 2-hydroxycinnam acid < quercetin < kaempferol < apigenin. This is in agreement with standards of gallic acid (6.0) < caffeic acid (12.5) < mangiferin (13.1) < trans-ferulic acid (14.6) < trans 3-hydroxycinnam acid (15.9) < 2-hydroxycinnam acid (15.6) < quercetin (17.5) < kaempferol (18.7) < apigenin (19.1). On the other hand, area for the compounds from least to highest were trans ferulic acid < caffeic acid < gallic acid < mangiferin < 2-hydroxycinnam acid < kaempferol < apigenin < quercetin < trans 3-hydroxycinnam acid (Figure 4).

### 2.4. Antimicrobial Activities of Heterotrigona itama Bee Bread

For this assay, BBE was used since it appeared to have the best antioxidant activity. The absorbance of various concentrations of *Klebsilla pneumonia, E. coli, Shigella* and *Salmonella typhi* and the positive controls (tetracycline, amoxycillin and Erythromycin) (Figure 5a–d) were normalized and used to obtain MIC_50_ values for antimicrobial activity. The antimicrobial activity of BBE was strongest in *Shigella* (MIC_50_: 1.617 µg/mL), followed by *Salmonella typhi* (MIC_50_: 1.813 µg/mL), *E. coli* (MIC_50_: 1.914 µg/mL) and *Klebsilla pneumonia* (MIC_50_: 1.923 µg/mL) (Figure 6a–f).

## 3. Discussion

Our previous studies assessed different samples of bee bread from three regions of Malaysia, namely: Selangor, Perak and Kelantan [21]. Results from the study revealed that bee bread from these regions possessed a variety of essential and non-essential amino acids, fats, carbohydrates, phenols and flavonoids, which gives it a high nutritive value. Further studies also revealed the presence of vitamins (B1, B2, A, E), calcium, iron, copper, sodium, potassium and cadmium; however, zinc and lead were not dictated in bee bread samples [22]. In the present study, we assessed the pH of bee bread, phytochemical screening, and in vitro antioxidant properties of three extracts of *Heterotrigona itama* bee bread (BBE, BBW and BBH). We also assessed the phenolic compounds analysis of BBE using HPLC, as well as the antimicrobial activity of the extract.

In this study, the pH of bee bread in the present study (pH was lower than the range previously reported. This may be a result of the activities of lactic acid bacteria during the fermentation of bee pollen to bee bread. Previous studies revealed that the pH of bee bread ranges between 3.8 and 4.3, which makes it differ from pH of bee pollen that ranges between 4.1 and 5.9. The low pH may be attributed to its high amount of lactic acid and carbohydrate, as well as low amount of proteins and fats as composition of bee bread may vary due to different sources and geographical locations [23,24].

There are several non-nutritive chemicals found in plants and natural products like honey products. These chemicals usually possess health-promoting properties. For many years, phytochemicals with unknown pharmacological activities have been studied as sources of phytotherapy. Research on phytochemicals has generally been considered as an effective approach in the discovery of new anti-infective agents from natural products. The samples of ethanolic extract of bee bread were used to determine preliminary phytochemical investigations which showed high concentrations of phenols, glycosides, alkaloids, resins, saponins, terpenoids, flavonoids, tannins and xanthoproteins. This is in line with studies carried out by Othman et al. [21], who reported that phytochemical compounds like xanthoproteins, flavonoids, phenols, saponins, tannins, terpenoids alkaloids, glycosides, resins, were found in bee bread ethanol extracts, while alkaloids were absent in bee bread aqueous extracts. The phytochemicals detected in this study are also similar to those reported previously [25], which have exhibited biological effects such as antimicrobial, antitumour and antihelmintic activities. Natural products usually possess large amounts of phenolic compounds, whose major role is antioxidant activities due to their redox properties, which makes them effective reducing agents, H^+^ donors and O_2_ extinguishers. To increase the nutritional quality of food, phenolic compounds assist in the oxidative degradation of lipids, making them essential in the food industries.

The present study revealed the presence of phenolic and flavonoid contents in BBE, corroborating previous studies. For example, ethanolic and hexanic extracts of *Heterotrigona itama* bee bread were reported to show antioxidant properties [26]. Similarly, a study by Othman et al. [21] carried out on three samples of Malaysian bee bread from three different regions, Kelantan, Perak and Selangor, showed significantly higher antioxidant phenol property and higher phenolic and flavonoid content in ethanol extract of bee bread compared to aqueous extract of *Heterotrigona itama* bee bread. Phenolic-rich natural products have higher phenolic contents and exhibit stronger antioxidant capacity than natural products with less phenolic contents. Therefore, natural products with higher amounts of phenolic compounds, carbohydrates, protein, lipid and ash are considered good food materials with significant health benefits.

*Heterotrigona itama* bee bread is rich in amino acids, minerals, as well as polyphenols, carotenoids, flavonoids, phytosterols and other compounds [13]. In this present study, HPLC analysis of BBE revealed nine compounds such as gallic acid, caffeic acid, mangiferin, trans ferulic acid, trans 3-hydroxycinnamic acid, trans 2-hydroxycinnamic acid, quercetin, kaempferol and apigenin. The amount of apigenin was the highest, followed by kaempferol. This is consistent with previous studies of Baltrušaitytė et al. [27], who reported that HPLC analysis of bee bread revealed high amounts of kaempferol, and a trace amount of p-coumaric acid, apigenin and chrysin, compared to honey. In addition to the compounds enumerated above, GC-MS results revealed trace quantities of flavonoids naringenin, quercetin, ferulic and caffeic acids in bee bread [28].

In a related study carried out on bee bread and bee pollen from Georgia, HPLC results identified the presence of quercetin, rutin and naringin [29]. A study carried out on six bee bread samples revealed thirty-two flavonoid derivatives, mainly kaempferol, quercetin, isorhamnetin, myricetin, and herbacetrin glycoside derivatives, with isrohamnetin-*O*-hexosyl-*O*-rutinoside, isorhamnetin-*O-*pentosyl-hexoside and quercetin-3-*O*-rhamnoside relatively abundant [30]. Samples of bee bread from Romania and India showed high amounts of kaempferol-3-*O-*glycosides and hydrocinnamic acid derivatives analyzed with High-Performance Liquid Chromatography with Diode Array Detection (HPLC/DAD) [31,32].

Previous study on samples of bee bread tested for antibacterial activities using *Pseudomonas aeruginosa*, *Staphylococcus aureus*, *Bacillus cereus* and *Escherichia coli* revealed that bee bread extracts were more sensitive to Gram-positive bacteria than Gram-negative bacteria [13]. In addition, antimicrobial activities of five samples of bee bread studied in Ukraine revealed the minimum inhibition concentration against the Gram-negative bacteria *Salmonella enterica* and *E. coli* as 6.40 μg/mL [33]. In Poland, alcoholic extracts of bee bread showed more antibacterial activity against Gram-positive bacteria when compared to water extract [34]. Ethanolic extract of bee bread inhibited the growth of *Streptococcus*; the bee bread showed inhibition against Gram-positive bacteria but not Gram-negative bacteria [27]. In our study, BBE inhibited the growth of microbes, with a higher activity against *Shigella* (MIC_50_: 1.617 µg/mL) than *Salmonella typhi* (MIC_50_: 1.813 µg/mL), *E. coli* (MIC_50_: 1.914 µg/mL) and *Klebsilla pneumonia* (MIC_50_: 1.923 µg/mL). This effect may be attributed to the rich polyphenol content of BBE, since polyphenols have significant antimicrobial properties.

## 4. Materials and Methods

### 4.1. Collection and Preparation of Heterotrigona itama Bee Bread

Bee bread samples were collected from a stingless bee farm (Mentari Technobee PLT), Kota Bharu, Kelantan, on the east coast region of Malaysia, Southeast Asia. During the dry season between the months of January and March, samples of *Heterotrigona itama* bee bread were collected and brought to the Laboratory of Department of Physiology, Universiti Sains Malaysia, Kelantan, Malaysia. They were weighed (initial weight) and then dried using a food dehydrator for 4 h at 35 °C to remove moisture. The samples were weighed again after drying (final weight). The blender was used to blend the sample into fine powder and stored at −20 °C until further analysis.

### 4.2. Preparation of Heterotrigona itama Bee Bread

To 50 g of BBE, BBW and BBH powder, 500 mL of 70% ethanol, water and hot water were added, respectively, stirred and kept for 72 h at room temperature, after which the mixtures were stirred at 500 rpm for 10 min. Thereafter, they were centrifuged for 10 min at 4000 rpm at 20 °C. Whatman filter paper No. 3 and No. 1 were used to filter the supernatant. The supernatants were freeze-dried and the final products (BBE, BBW and BBH) were collected for analysis.

### 4.3. pH of Heterotrigona itama Bee Bread

Three hundred mg of each *Heterotrigona itama* bee bread sample were used to determine the pH. Briefly, each bee bread sample was dissolved in a 1.5 mL centrifuge tube containing 300 μL of distilled water or ethanol. A pH meter with an accuracy of +0.01 was used to measure the pH of the mixture.

### 4.4. Phytochemical Screening of Heterotrigona itama Bee Bread

Qualitative phytochemical screening was carried out on *Heterotrigona itama* bee bread extracts to determine the presence of alkaloids, flavonoids, phenols, tannins, saponins, terpenoids, resins, glycosides and xanthoproteins.

The presence of alkaloids was evaluated using Mayer’s test, as previously described [35]. Briefly, 5 mL of 2% (*v/v*) of HCl were added to 50 mg of extract, stirred and filtered. Thereafter, 1 mL of Mayer’s reagent was added to the filtrate. A cream colour indicated the presence of alkaloids.

The presence of flavonoids was tested using a previously described method [36]. Next, 200 mg of extract were placed in a test tube, followed by the addition of 10% ferric chloride. A brownish colour indicated the presence of flavonoid. In the test for phenols, few drops of 5% ferric chloride were added to a test tube containing 50 mg of bee bread. A deep purple colour indicated phenols presence [35]. In the test for tannins, on the other hand, 2–3 drops of 0.1% ferric chloride were added to a test tube containing 50 mg of extract and warmed for some minutes. The appearance of olive-green colour indicated the presence of tannins [37].

To test for the presence of saponin, three drops of olive oil were added to a test tube containing 500 mg of extract. The persistent of froth and emulsion confirmed the presence of saponin [37]. To test for terpenoids, 2 mL of chloroform and 3 mL of 98% (*v/v*) H_2_SO_4_ were pipetted into a test tube, followed by the addition of 500 mg of extract. The appearance of a reddish-brown coloured layer indicated the presence of terpenoid [37].

To establish of presence of glycoside, a drop of 10% ferric chloride and 1 mL of concentrated sulphuric acid were added to a test tube containing 1 mL of glacial acetic acid. Thereafter, 100 mg of extract were added along the side of the test tube. A reddish-brown colour confirmed the presence of glycoside [37]. To test for xanthoproteins, 200 mg of extract were placed in a test tube, and 3 drops of 15% (*v/v*) nitric acid were added. A yellow colour indicated the presence of xanthoproteins [37].

### 4.5. In Vitro Antioxidant Property of Heterotrigona itama Bee Bread

In vitro antioxidant properties of the extracts were carried out to determine the DPPH radical scavenging activity, FRAP, H_2_O_2_ scavenging activity, total phenolic and flavonoid contents.

#### 4.5.1. DPPH Radical Scavenging Activity

The method of Oršolić et al. [38] was used to determine DPPH radical scavenging activity of the extracts. Briefly, 1000 μL of extract or standard prepared in methanol were added to DPPH methanol solution (0.16 mM). The mixture was vortexed and kept in the dark for 30 min at room temperature, after which the absorbance was read at 512 nm wavelength using a spectrophotometer. Methanol (1000 μL) was used as a control. The DPPH radical scavenging activity was performed in triplicates and the results were computed using the equation below:

DPPH radical scavenging activity (%) = [(ODc–ODs)/ ODc] × 100

Where: ODc = absorbance of DPPH with methanol (control)

ODs = absorbance of DPPH with extract or standard.

#### 4.5.2. Ferric Reducing Antioxidant Power

The method as described by Benzie and Strain [39] was used to assess the FRAP activity of the extract. The method is based on the reduction of ferric tripyridyltrazine complex to ferrous tripyridyltrazine. Briefly, 200 μL of bee bread extract were added to a test tube containing 1500 μL of freshly prepared FRAP reagent made up of 10 mM 2,4,6-tripyridyl-S-triazine in 40 mM HCl, 20 mM FeCl_3_ solution and 0.3 M acetate buffer in a ratio of 1:1:10, then the mixture was incubated for 4 min at 37 °C, after which the absorbance was read spectrophotometrically at 593 nm wavelength against distilled water blank. This assay was conducted in triplicates and results expressed in millimoles of ferrous per litre (mmol Fe^2+^ Eq L^−1^).

#### 4.5.3. Hydrogen Peroxide Scavenging Activity

The H_2_O_2_ scavenging activity of bee bread extract was assessed according to a previously described method [40]. Concisely, to a test tube containing 0.6 mL of H_2_O_2_ (40 mM) solution in 0.1 M phosphate buffer (pH 7.4), 3.4 mL of the extract or standard prepared in 0.1 M phosphate buffer (pH 7.4) were added and mixed thoroughly. Phosphate buffer was used as blank. Thereafter, the mixture was incubated for 10 min. Subsequently, the absorbance of the mixture was read spectrophotometrically at 230 nm wavelength. The H_2_O_2_ scavenging activity was performed in triplicates and results calculated and expressed in percentage using the equation:

H_2_O_2_ scavenging activity (%) = [(ODc–ODs)/ODc] × 100

Where: ODc = absorbance of blank

ODs = absorbance of sample

#### 4.5.4. Total Phenolic Content

The method of Folin-Ciocalteu [41] was used to determine the total phenolic content in the extracts. Briefly, 200 μL of each bee bread extract or standard were added into their respective test tubes containing 1 mL Folin-Ciocalteu phenol reagent in a ratio of 1:10, mixed and allowed to stand for 3 min. Then, 1 mL of 10% sodium bicarbonate was added to the mixture and allowed to stand in the dark for 90 min after being made up to 10 mL with distilled water. The absorbance was read spectrophotometrically at 725 nm wavelength. Total phenolic content was determined from a standard curve generated using gallic acid (20, 40, 60, 80 and 100 μg/mL). The assay was conducted in triplicates and results expressed in milligrams gallic acid equivalent per gram of extract (mg GAE g^−1^).

#### 4.5.5. Total Flavonoid Content

The method described by El Hariri et al. [42,43] was used to determine the TFC in the extract. The reaction mixture contained bread extract (1 mL), 4 mL of 70% ethanol, 300 μL of sodium nitrite (5 % *w/v*), 300 μL of aluminum chloride (10% *w/v*), 2 mL of 1 M sodium hydroxide and distilled water was added to make up the volume to 10 mL. Absorbance was read at 510 nm wavelength after the mixture was properly vortexed. TFC was calculated from a standard curve generated using quercetin (20, 40, 60, 80 and 100 μg/mL). The TFC assay was conducted in triplicates and the results calculated were expressed in milligrams quercetin equivalent per gram of extract (mg QE g^−1^).

### 4.6. Determination of the Antimicrobial Activity of Heterotrigona itama Bee Bread

#### 4.6.1. Minimum Inhibitory Concentration (MIC) Determination

The microdilution method was used to determine the antimicrobial activity, employing Mueller-Hinton broth and Luria Bertania (LB) principles. BBE was dissolved in dimethyl sufoxide (DMSO, 10% of the final volume) and diluted with culture broth to a concentration of 2 mg/mL. Furthermore, a concentration ranging from 2 to 0.0156 mg/mL was attained by adding culture broth through 1:2 serial dilution, out of which 100 μL of the dilutions were pipetted into 96-well plates, as well as a sterility control and a growth control. Each test and growth control well were inoculated with 5 μL of a bacterial suspension (108 CFU/mL or 105 CFU/well). All experiments were done in triplicate, after which the microdilution trays were incubated at 36 °C for 18 h. Bacterial growth was detected firstly by optical density (ELISA reader, CLX800-BioTek, Winooski, VT, USA) and by the addition of 20 μL of an INT alcoholic solution (0.5 mg/mL) (Sigma, St. Louis, MO, USA). The trays were again incubated at 36 °C for 30 min, and in those wells where bacterial growth occurred, INT changed from yellow to purple. MIC values were defined as the lowest concentration of each natural product, which completely inhibited microbial growth. The results were expressed in mg/mL.

#### 4.6.2. Disc Diffusion Assay

The antimicrobial activity of bee bread was performed by the disc diffusion method. Briefly, 0.5 McFarland standards (1 × 108 CFU/mL) was prepared overnight on cultures of bacteria (*K. pnemonia, Shigella, E. coli and S. typhi*) using peptone water; 10 μL of each suspension was spread on the solid nutrient agar plates. A sterile disc of 6 mm was loaded with 10 μL of BBE, antibiotics (ampicillin, tetracycline, amoxycillin) and DMSO as positive and negative controls, respectively. The sterile discs were then placed onto the surface of the inoculated nutrient agar plates. Thereafter, agar plates were incubated for 24 h at 37 °C. The diameter of the inhibition zones was measured in millimetres (mm).

#### 4.6.3. Broth Dilution Method

The minimum inhibitory concentration (MIC) of BBE was determined as described by Chen et al. [44]. The 10 mL sterile nutrient broth adjusted to appropriate turbidity was used to grow isolated colonies of the selected bacteria. Thereafter, it was cultured in various concentrations of BBE for 24 h. The UV-Vis spectrophotometer (Shanghai Spectrum Instruments Co Ltd. Shanghai, China) was used to measure the initial turbidity of culture at 600 nm wavelength while the MIC of BBE was determined based on the cultures that were incubated at the same wavelength. MIC values were defined as the lowest concentration of each bee bread, which completely inhibited microbial growth. The experiment was performed in triplicates.

### 4.7. High-Performance Liquid Chromatography (HPLC) Analysis of Heterotrigona itama Bee Bread

Bee bread was screened for the presence of phenolic and flavoinoid compounds using HPLC technique. Briefly, the chromatography system with a flow rate of 1.0 mL/min, a Zorbax SB-C18 column (3.5 µm, 4.6 mm I.D × 150 mm) at an oven temperature of 25 °C, a mobile phase of 0.1% formic acid in water and 0.1% formic acid in methanol (40:60, *v/v*), and a detection wavelength of 340 nm were used to carry out the HPLC assay. A volume of 20 μL of sample was then injected into it. Thereafter, the solutions were filtered through a 0.45 μm nylon membrane prior to HPLC injection. The chromatographic analyses were conducted at a run time of 0, 20, 25, 25.1 and 30 min.

### 4.8. Statistical Analysis

GraphPad Prism 7.0 for windows (GraphPad Software Inc., LaJolla, CA, USA) was used to analyse the data. One-way analysis of variance (ANOVA) followed by Tukey’s post hoc test was used to analyse values with normal distribution and homogenous variance. Data are presented as mean ± standard deviation (SD). *p*-value < 0.05 was considered statistically significant.

## 5. Conclusions

Our study revealed that BBE showed significantly higher DPPH radical scavenging activity, H_2_O_2_ scavenging activity, total phenolic and flavonoid contents as well as FRAP activity, than BBW and BBH. HPLC analysis of BBE showed the presence of five phenolic acids (gallic acid, caffeic acid, trans-ferulic acid, trans 3-hydroxycinnamic acid, 2-hydroxycinnamic acid) and four flavonoids (quercetin, kaempferol, apigenin and mangiferin). Finally, the antimicrobial activity of BBE was stronger in *Shigella*, followed by *Salmonella typhi*, *E. coli* and *Klebsilla pneumonia.* Bee bread used in the present study, therefore, possesses compounds (flavonoids and phenolic acids) which can reduce oxidative stress, inflammation and apoptosis, and prevent microbial growth, thereby making it a promising therapy for some disease conditions.

## Figures and Tables

**Figure 1 molecules-26-04943-f001:**
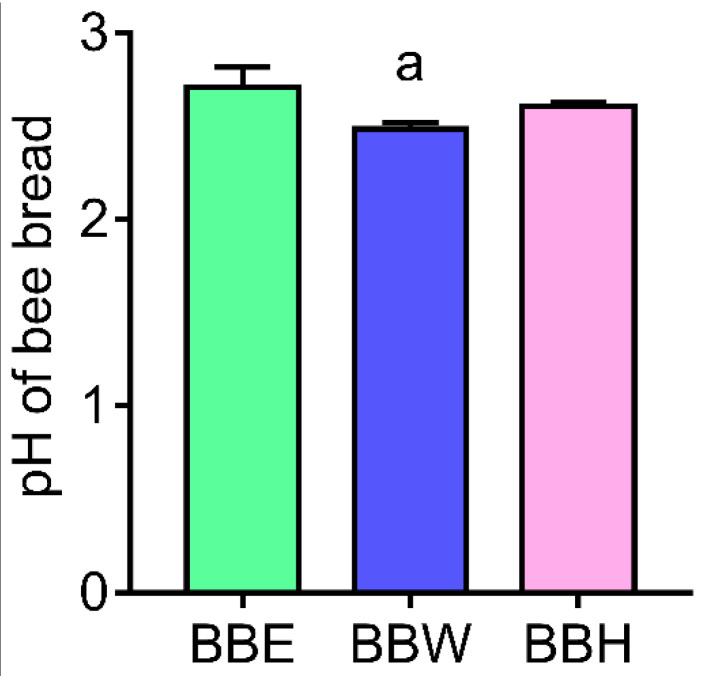
pH of bee bread. BBE: bee bread ethanol extract, BBW: bee bread water extract. BBH: bee bread hot water extract, ^a^
*p* < 0.05 versus BBE.

**Figure 2 molecules-26-04943-f002:**
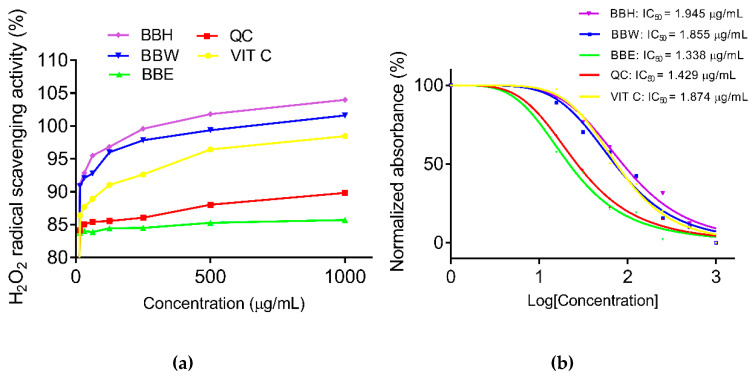
H_2_O_2_ scavenging activity (**a**) and normalized absorbance of H_2_O_2_ (**b**). H_2_O_2_: hydrogen peroxide, BBE: bee bread ethanol extract (IC_50_: 1.338 µg/mL), BBW: bee bread water extract (IC_50_: 1.855 µg/mL), BBH: bee bread hot water extract (IC_50_: 1.945 µg/mL), VITC: Vitamin C (IC_50_: 1.874 µg/mL), QC: quercetin (IC_50_: 1.429 µg/mL).

**Figure 3 molecules-26-04943-f003:**
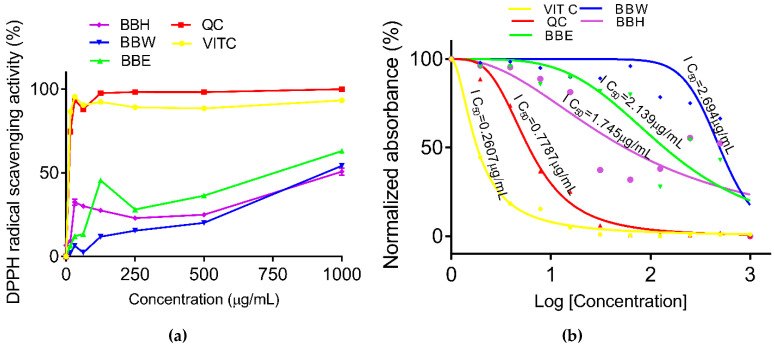
DPPH radical scavenging activity (**a**) and normalized absorbance of DPPH activity (**b**). DPPH: 1,1 diphenyl 2 picrylhydrazyl, BBE: bee bread ethanol extract (IC_50_: 2.139 µg/mL), BBW: bee bread water extract (IC_50_: 2.694 µg/mL), BBH: bee bread hot water extract (IC_50_: 1.745 µg/mL), VITC: Vitamin C (IC_50_: 0.2607 µg/mL), QC: quercetin (IC_50_: 0.7787 µg/mL).

**Figure 4 molecules-26-04943-f004:**
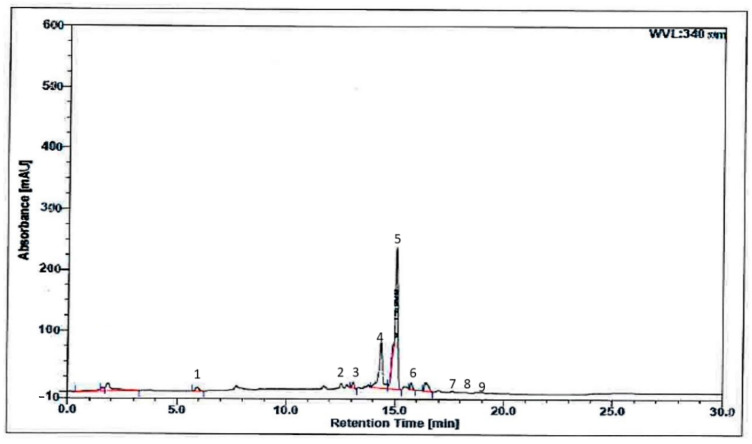
High-performance liquid chromatography analysis of BBE at 340 nm. Peak 1: gallic acid, peak 2: caffeic acid, peak 3: mangiferin, peak 4: trans-ferulic acid, peak 5: trans 3-hydroxycinnam acid, peak 6: 2-hydroxycinnam acid, peak 7: quercetin, 8: kaempferol, peak 9: apigenin.

**Figure 5 molecules-26-04943-f005:**
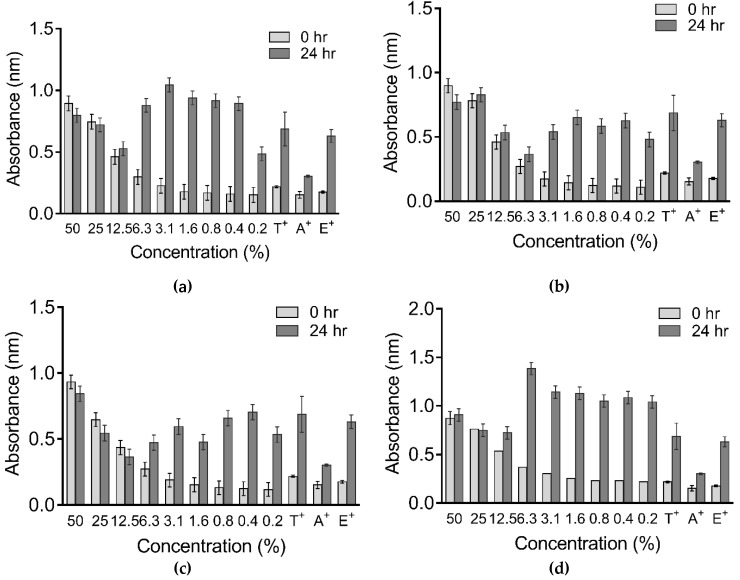
Absorbance of various concentrations of (**a**): *E. coli*, (**b**) *S. typhi*, (**c**): *Shigella* and (**d**) *K. pneumonia.* The absorbance was measured at 0 h and after 24 h of incubation. T: tetracycline, A: amoxycillin, and E: erythromycin were used as positive controls.

**Figure 6 molecules-26-04943-f006:**
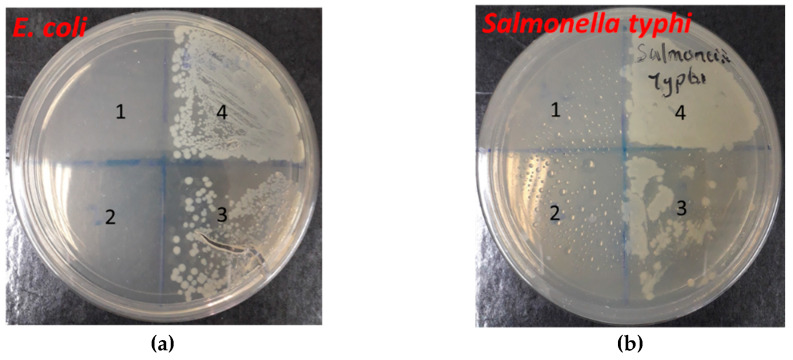
Antimicrobial activity of BBE in (**a**) *E. coli*, (**b**) *Salmonella typhi*, (**c**) *Shigella*, (**d**) *K. pneumonia,* (**e**) Erythromycin (control), (**f**) normalized absorbance of *E. coli*, *Salmonella typhi*, *Shigella* and *K. pneumonia* from which the MIC_50_ were calculated.

**Table 1 molecules-26-04943-t001:** Phytochemical screening analysis of BBE, BBW and BBH extract.

Compound	BBE	BBW	BBH
Phenols	++	+	+
Glycosides	++	+	+
Alkaloids	++	+	+
Xanthoproteins	+	+	+
Terpenoids	+++	+	++
Tannins	+	+	+
Resins	++	+	++
Saponins	++	++	++
Flavonoids	+++	+	++

BBE: bee bread ethanol extract, BBW: bee bread water extract: BBH: bee bread hot water extract, Positive sign (+) represents the intensity of colour change. The higher the concentration the more the positive sign.

**Table 2 molecules-26-04943-t002:** In vitro antioxidant activities of BBE, BBW and BBH extracts.

Parameter	BBE	BBW	BBH
DPPH radical scavenging activity (%)	85.79 ± 0.40	7.62 ± 0.13 ^a^	8.47 ± 0.01 ^a,b^
Hydrogen peroxide scavenging activity (%)	90.53 ± 2.14	74.21 ± 1.61	69.61 ± 0.98 ^a,b^
FRAP (µmol Fe^2+^ Eq/L)	106.7 ± 1.47	68.49 ± 0.94 ^a^	48.59 ± 1.95 ^a,b^
Total phenolic content (mg GAE g^−1^)	17.44 ± 0.93	9.80 ± 0.10 ^a^	9.55 ± 1.00 ^a^
Total flavonoid content (mg QE g^−1^)	21.34 ± 0.83	2.69 ± 0.09 ^a^	4.82 ± 0.37 ^a,b^

BBE: bee bread ethanol extract, BBW: bee bread water extract: BBH: bee bread hot water extract, DPPH: 1,1 diphenyl 2 picrylhydrazyl, FRAP: ferric ion reducing antioxidant power. Values are expressed in mean ± SD, n = 3. ^a^ *p* < 0.05 versus BBE, ^b^
*p* < 0.05 versus BBW (one-way analysis of variance followed by Tukey’s post hoc test).

## Data Availability

The datasets for this manuscript can be obtained from the corresponding author upon reasonable request.

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
