# Peer review of "Chemical Profile, Antioxidant Properties and Antimicrobial Activities of Malaysian Heterotrigona itama Bee Bread"

_molecules, 2021, doi:10.3390/molecules26164943_

Round 1
Reviewer 1 Report
Article
Phytochemical screening, antioxidant properties, high performance liquid chromatography and antimicrobial activities of Malaysian Heterotrigona itama Bee bread
The article is interesting, since few bibliographies were found, the information provided by this article is complete and relevant, but:
It would be necessary to evaluate and improve article’ title
Evaluate English throughout the article, is poor and in some points worrying
Some errors of which I will cite two of them
Line 41 Samonella typhimurium
Line 42 microbial properties? / antimicrobial properties
Author Response
Reviewer #1
Reviewer comments:
The article is interesting, since few bibliographies were found, the information provided by this article is complete and relevant.
Authors’ response
Thank you for your kind comments.
Reviewer comments:
It would be necessary to evaluate and improve article’ title
Authors’ response
Thank you for your kind comments. The title has been revisited.
Reviewer comments:
Evaluate English throughout the article, is poor and in some points worrying
Authors’ response
The English language used in the article has been improved.
Reviewer comments:
Some errors of which I will cite two of them
Line 41 Samonella typhimurium
Authors’ response
We apologise for this error. It has been corrected.
Reviewer comments:
Line 42 microbial properties? / antimicrobial properties
Authors’ response
We apologise for this error. It has been corrected. The entire manuscript has been scrutinised for errors.
Reviewer 2 Report
In this work, Authors evaluated the chemical profile, antioxidant effects and antimicrobial activity of Heterotrigona itama bee bread from Malaysia. The article could be interesting for Molecules readers but it needs major revision according to the comments and recommendations given below:
1) In the paragraph "2.4 Phytochemical screening of Heterotrigona itama bee bread" the description of all methods must be improved.
2) The paragraph "2.7 High performance liquid chromatography (HPLC) analysis of Heterotrigona itama bee bread" must be improved.
3) The quality of Figure 1 must be improved.
4) In the biological tests, data obtained with the positive control must be reported.
5) The quantitative determination of constituents must be inserted.
Author Response
Reviewer #2
Reviewer comments:
In this work, Authors evaluated the chemical profile, antioxidant effects and antimicrobial activity of Heterotrigona itama bee bread from Malaysia. The article could be interesting for Molecules readers but it needs major revision according to the comments and recommendations given below:
Authors’ response
Thank you for your kind comments.
Reviewer comments:
1) In the paragraph "2.4 Phytochemical screening of Heterotrigona itama bee bread" the description of all methods must be improved.
Authors’ response
The descriptions of all the methods have now been improved (Now in 4.4).
Reviewer comments:
2) The paragraph "2.7 High performance liquid chromatography (HPLC) analysis of Heterotrigona itama bee bread" must be improved.
Authors’ response
The HPLC of bee bread has now been improved (Now in 4.7).
Reviewer comments:
3) The quality of Figure 1 must be improved.
Authors’ response
The figure has been edited and improved accordingly.
Reviewer comments:
4) In the biological tests, data obtained with the positive control must be reported.
Authors’ response
The positive control has now been reported.
Reviewer comments:
5) The quantitative determination of constituents must be inserted.
Authors’ response
Thank you for your kind comment. We quantified total phenols and total flavonoids.
Reviewer 3 Report
This is very nice experimental work concerning biological properties of one of the natural products made by honey bees. Both experiments and discussion of the results are comprehensive, clear and edequately described (i.e. objectively and not exagerating results). Authors provided also adequate literature background.
I have noticed only several editorial mistakes, e.g. Staphlococcus aureus and Staphlococcus epidermidis (line 87),
Another issue is Table 1,showing phytochemical screening analysis, where according to the footnote one or two pluses represent: "Positive sign (+) represents the intensity of colour change. The deeper the concentration the more the positive sign"
I am aware that this approach is used in mamy laboratories. However, is this possible to quantify this result? Please comment.
Anyway, this is intersting work and should be published with minor revision.
Author Response
Reviewer #3
Reviewer comments:
This is very nice experimental work concerning biological properties of one of the natural products made by honey bees. Both experiments and discussion of the results are comprehensive, clear and adequately described (i.e. objectively and not exagerating results). Authors provided also adequate literature background.
Authors’ response
Thank you for your kind comments.
Reviewer comments:
I have noticed only several editorial mistakes, e.g. Staphlococcus aureus and Staphlococcus epidermidis (line 87),
Authors’ response
These mistakes have been corrected.
Reviewer comments:
Another issue is Table 1, showing phytochemical screening analysis, where according to the footnote one or two pluses represent: "Positive sign (+) represents the intensity of colour change. The deeper the concentration the more the positive sign"
I am aware that this approach is used in many laboratories. However, is this possible to quantify this result? Please comment.
Authors’ response
Thank you for your kind comments. However, quantification of phytochemicals is possible but in this present study we only concentrated on the qualitative analysis.
Reviewer comments:
Anyway, this is interesting work and should be published with minor revision.
Authors’ response
Thank you for your kind comments.
Round 2
Reviewer 2 Report
This work can be accepted for publication in this revised form.